# The role of resource transfer in positive, non-additive litter decomposition

**Na Yin, Roger T. Koide***

Department of Biology, Brigham Young University, Provo, Utah, United States of America

* rogerkoide@byu.edu

**Data Availability Statement:** All relevant data are within the paper and its Supporting Information files.

**Funding:** We gratefully acknowledge the financial support provided by Brigham Young University to RTK, the Office of Research and Creative Activities,

## Abstract

Naturally occurring, mixed litter decomposes at unpredictable rates when individual components do not decompose in mixtures as they do individually. Consequently, nutrient, carbon and energy fluxes associated with decomposition may be difficult to predict. However, predictability is improved when we understand the mechanisms responsible for such non-additive decomposition. In this study, we explored mechanisms to explain our previous observation that an approximately 30% increase in oat straw decomposition due to the presence of clover litter is associated with a significant increase in the activity of cellobiohydrolase, an enzyme involved in litter decomposition. We hypothesized that resources limiting decomposer microbe enzyme activity in oat straw can be supplied by clover litter. Amendment of oat straw with water, $NH_4Cl$, glucose, or $NH_4Cl$ combined with glucose did not account for the significant, positive effect of clover litter on oat straw decomposition and cellobiohydrolase activity. However, amendment of oat straw with a complete set of mineral nutrients for plant growth did account for the entire effect of clover litter, and the addition of the complete set of mineral nutrients without N accounted for the majority of the clover effect. In our system, therefore, the majority of the positive effect of clover litter on oat straw decomposition and cellobiohydrolase activity was unexpectedly not attributable to the transfer from clover to oat straw of labile N. We found that mineral soil could substitute for the mineral nutrients other than N. This highlights the role of soil as a potential source of limiting resources for microbes decomposing litter. It may also explain why positive, non-additive decomposition has been observed in some previous studies but not in others depending on whether the soil supplied a resource that limited decomposer activity.

## Introduction

A clear understanding of the controls of litter decomposition is of obvious importance. The transfer of energy in litter to the soil food web [1] requires litter decomposition, which results in the cycling of mineral nutrients [2], a flux of carbon (C) into the atmosphere, and the transformation of relatively labile litter into stable soil organic matter [3–5]. In both agricultural

Brigham Young University to NY, and by the Sustainable Bioenergy Research Program of the USDA National Institute of Food and Agriculture (# 2011-67009-20072) to RTK.

**Competing interests:** The authors have declared that no competing interests exist.

and natural ecosystems, litter usually occurs in heterogeneous mixtures comprising multiple decomposition stages and multiple plant species [6]. Moreover, new combinations of plant species are proliferating as plant community composition is altered by biological invasion, physical disturbance, climate change or novel intercropping strategies. Unfortunately, it is far more difficult to predict decomposition rates of heterogeneous mixtures of litter than of single litter types; one litter type may have significant positive or negative effects on the decomposition of another [7,8].

Cases of this "non-additive" decomposition of mixed litters are common [8–10], and the effect of one litter type on another may be quite large—from a 22% negative impact to a 65% positive impact [8]. For example, we previously showed that in an oat straw-clover litter system, positive, non-additive decomposition, characterized by an approximately 30% increase in oat straw decomposition due to the presence of clover litter, was associated with enhanced decomposition enzyme activity in the oat straw [11]. Decomposer fungi produce networks of hyphae connecting fragments of litter, and these networks may transfer limiting resources from resource-rich litter to resource-poor litter [6,12,13], increasing the metabolic activity of decomposer microbes in resource-poor litter and accelerating its decomposition. We, therefore, now address the mechanisms leading to positive, non-additive decomposition of oat straw due to the presence of clover litter by determining the resources that limit oat straw decomposition enzyme activity.

Potentially-limiting resources include water, mineral nutrients and labile C. Litter moisture frequently limits the activity of decomposers [2]. Physical structure varies among litter types, and structure markedly affects the capacity to retain water [14]. Therefore, the transfer of water from one litter type to a second may influence its decomposition. Indeed, Wardle et al. [15] found a positive effect of litters of high water-holding capacity on the decomposition of litters of low water holding capacity. Stimulation of decomposer organisms may also occur as a consequence of the addition of labile sources of C [16,17] because litter composed primarily of recalcitrant C compounds may not meet the demand for energy by decomposer organisms. Carbon compounds in litter range from labile compounds such as carbohydrates and amino acids, to more recalcitrant compounds, such as condensed tannins and lignin, and there is a good correlation between decomposition rate and the initial concentration of labile C compounds [18]. Mineral nutrients frequently limit microbial activity during litter decomposition [19–22]. Positive, non-additive decomposition may occur in litter mixtures when the litter types differ markedly in N availability, suggesting an important role for N transfer between litter types [7,23,24]. In some cases, phosphorus (P), rather than N, may be the nutrient that most limits microbial litter decomposition [25,26]; Montané et al. [27] found that decomposition of P-poor litter increased when mixed with P-rich litter. In other studies, the transfers of potassium, magnesium, manganese, and calcium from one litter type to another were associated with increased decomposition of the latter [28].

Therefore, our goal was to determine whether specific resources that could be supplied to oat straw by cover litter limited decomposition enzyme activity in oat straw. Specifically, we hypothesized that water, labile C, labile N, other mineral nutrients, or combinations of these resources provided by clover litter stimulated microbial enzyme activity in oat straw and accounted for the significant positive effect of clover litter on oat straw decomposition. We chose oat straw and clover litter as our model litter mixture because both are common crops in the U.S.A., because legume–grass mixtures are common in many agricultural systems around the world, and because our previous study demonstrated a significantly positive, non-additive effect of clover on oat straw decomposition [11].

## Materials and methods

### Plant materials

Oat (*Avena sativa* L.) and red clover (*Trifolium pratense* L.), both planted in the fall of 2014, were harvested by hand on 19 May 2015 from an experimental farm at the Pennsylvania State University, State College, PA, USA (40˚43'17.70"N, 77˚55'39.87"W, approximately 365 m elevation). Grain had not been previously harvested from the oats, but the stems we collected were mostly leafless. Clover was harvested at mid-bloom. We did not harvest litter on the soil surface because it would already have started decaying, and clover decays very rapidly. Instead, we harvested standing but dead oat straw and live clover shoots, and used these materials in our decomposition experiments. Materials were relatively soil free but not washed. We hereafter refer to these materials as oat straw and clover litter, respectively. The field-collected materials were placed in paper bags and dried at 65˚ C, then maintained at room temperature at Brigham Young University, UT, USA. The initial litter total N concentration was analyzed with a Combustion—Elementar Vario Max N/C Analyzer. Initial litter P, K, Ca, Mg and S concentrations were determined by inductively-coupled plasma (ICP) emissions spectrometry following digestion in nitric acid. All the measurements of litter nutrient quality were conducted at Agricultural Analytical Services Lab, Pennsylvania State University, State College, PA, USA.

### Common methods for Experiments 1, 2, 3.1, 3.2, 4, 5, 6

**Mesh bags.** Mesh bags (7 x 8 cm) were constructed from window screen material (PVC-coated nylon mesh, 1.5 mm mesh size) using a heat sealer. The bags were used to contain either litter or fiberglass pads, which were used to hold water or aqueous solutions of various composition depending on the experiment, see below. In order to easily document oat straw decomposition, oat straw and clover litter decompositions were carried out in separate bags, which could either be incubated separately or in combination, see below.

**Oat straw or clover litter.** Dried oat straw was cut into 1.5 cm length pieces and mixed thoroughly. Clover litter was mixed thoroughly and was small enough not to require further cutting. Mesh bags were each filled with 1±0.05 g of oat straw or clover litter. Mesh bags containing litter were then saturated in distilled water for 4 h and then allowed to drain, resulting in oat straw or clover litter at field capacity. Approximately 3.0 or 3.5 g water remained in each 1 g oat straw or 1 g clover litter, respectively.

**Treatments with water or various solutions.** Water, $NH_4Cl$ solution, glucose solution or a solution of other mineral nutrients were supplied to oat straw via an accompanying mesh bag containing an inert, fiberglass pad (6.5 x 7.5 cm x 0.25 cm thick). For experiments 1, 2, and 3.1, we pipetted 3.5 mL purified water or appropriate nutrient solution once at the beginning of the experiment onto each fiberglass pad, approximately the same amount of water held by a 1 g sample of dry clover litter saturated for 4 hours and then allowed to drain freely. For experiments 3.2, 4, 5, 6, 7 and 8, we pipetted 3.5 mL purified water or appropriate nutrient solution at the beginning of the experiment onto each fiberglass pad to supply the listed amounts of resources (see individual experiments, below) and, in subsequent weekly intervals (7, 14 and 21 d after the start), added additional 0.4 mL water or solution aliquots, each containing the same amount of nutrient in the original 3.5 mL, to the fiberglass pads.

**Arrangement of mesh bags.** Oat straw mesh bags were incubated either by themselves or with a mesh bag containing either clover litter or a fiberglass pad containing water or appropriate nutrient solution (see below). When an oat straw mesh bag was incubated with another mesh bag, the oat straw mesh bag was stapled to and on top of the other mesh bag in order to prevent liquids moving by gravity from the other mesh bag to the oat straw. Therefore,

movement against gravity would have to be accomplished by capillary action or hyphal translocation.

**Incubation conditions during decomposition.** Each replicate mesh bag or replicate mesh bag set was incubated separately in its own petri dish, and all petri dishes were placed in one, enclosed plastic container. A higher than ambient humidity (unknown value) was maintained within the container with damp paper towels placed in the bottom. The container was kept in the laboratory at constant temperature, $23 \pm 0.5°$ C.

**Calculation of oat straw decomposition rate.** Intact mesh bags were oven-dried at $65°$ C and litter samples were weighed. Decomposition rate was calculated as the difference between the original and final dry weights, divided by the intervening time period. When subsamples were taken for both the calculation of decomposition rate and enzyme activity (see below), total sample wet weight, subsample wet weights and subsample dry weights were used in the calculations.

**Oat straw cellobiohydrolase activity.** Cellobiohydrolase (CBH, EC 3.2.1.91) is one of the major enzymes involved in cellulose hydrolysis. Where indicated for individual experiments, subsamples of oat straw were ground to a fine powder using mortar and pestle in liquid nitrogen, and CBH activity was measured fluorometrically using the substrate 4-methylumbelliferyl-β-D-cellobioside (MUB-CB, P212121), a method modified from Peoples and Koide [29]. Full details are given in S1 Appendix.

## Experiment 1. Amendment with water

The purpose of this experiment was to determine whether the positive effect of clover litter on oat straw decomposition could be attributed to water supplied by the clover litter. There were five replicates of each of the two treatments: 1) oat straw, and 2) oat straw with an additional source of water to mimic the water supplied by clover litter. Litters were harvested after 25 days of incubation. Incubation times were determined from preliminary experiments. Decomposition rates of oat straw samples were calculated as explained above.

## Experiment 2. Amendment with NH₄Cl

The purpose of this experiment was to determine whether the positive effect of clover litter on oat straw decomposition could be attributed to N supplied by the clover litter. There were five replicates each of four treatments: 1) oat straw with clover litter, 2) oat straw with additional water, 3) oat straw with a solution containing 2000 μg N as $NH_4Cl\ g^{-1}$ dry weight of litter (equal to 25% of the total N in a 1 g oat straw sample, assuming a concentration of 0.8% N, data not shown), and 4) oat straw with a solution containing 4000 μg N as $NH_4Cl\ g^{-1}$ dry weight of litter (equal to 50% of the total N in a 1 g oat straw sample). Litters were harvested after 25 days of incubation. Decomposition rates of oat straw samples were calculated as explained above.

## Experiments 3.1 and 3.2. Amendment with glucose

**Experiment 3.1.** The purpose of this experiment was to determine whether there was an optimum amount of glucose to stimulate oat straw decomposition. We used cellobiohydrolase activity as a quick proxy for decomposition because a previous study indicated the two were correlated, as they also proved to be in this study (see Results). There were six replicates for each of seven treatments: 1) oat straw with additional water, 2) oat straw with 50 μg C as glucose $g^{-1}$ dry litter, 3) oat straw with 100 μg C as glucose $g^{-1}$, 4) oat straw with 200 μg C as glucose $g^{-1}$, 5) oat straw with 500 μg C as glucose $g^{-1}$, 6) oat straw with 1000 μg C as glucose $g^{-1}$, and 7) oat straw with 3500 μg C as glucose $g^{-1}$. Litter bags were harvested after 7 days of

incubation. Each litter sample was assessed for cellobiohydrolase activity as explained above and detailed in the S1 Appendix.

**Experiment 3.2.** The purpose of this experiment was to determine whether the optimum concentration of glucose (as determined in Experiment 3.1) could account for the positive effect of clover litter on oat straw decomposition. This experiment had four treatments: 1) oat straw with clover, 2) oat straw with additional water, 3) oat straw with 500 μg C as glucose $g^{-1}$ dry litter, and 4) oat straw with 4000 μg C as glucose $g^{-1}$. Each treatment was replicated 24 times so that 8 replicates of each treatment could be sampled on each of 3 occasions (at 14, 21, 28 d of incubation). Each litter sample was separated into two subsamples, one of which was used to calculate the rate of oat straw decomposition and the other to determine cellobiohydrolase activity as explained above and detailed in the S1 Appendix.

## Experiment 4. Simultaneous amendment with glucose and NH$_4$Cl

The purpose of this experiment was to determine whether the simultaneous addition of labile C and mineral N could account for the positive effect of clover litter on decomposition of oat straw. There were eight replicates of each of four treatments: 1) oat straw with clover, 2) oat straw with additional water, 3) oat straw with 500 μg C as glucose $g^{-1}$ dry litter, and 4) oat straw with 500 μg C as glucose $g^{-1}$ dry litter plus 137 μg N as NH$_4$Cl $g^{-1}$ dry litter. The C: N ratio of the latter mixture, 3.65, is equivalent to that of plant peptone, a hydrolysate of plant protein. Mesh bags were harvested after 28 days of incubation. Each litter sample was separated into two subsamples, one of which was used to calculate rate of oat straw decomposition, and the other to determine cellobiohydrolase activity as explained above and detailed in the S1 Appendix.

## Experiment 5. Simultaneous amendment with glucose, NH$_4$Cl and other mineral nutrients

The purpose of this experiment was to determine whether the simultaneous addition of glucose, NH$_4$Cl and other mineral nutrients could account for the effect of clover litter on decomposition of oat straw. There were eight replicates of each of three treatments: 1) oat straw with clover litter, 2) oat straw with additional water and 3) oat straw with 500 μg C as glucose $g^{-1}$ dry litter plus 137 μg N as NH$_4$Cl $g^{-1}$ dry litter and all other mineral nutrients supplied in the same ratio to N as in Hoagland solution [30]. For treatment 3, in addition to the NH$_4$Cl, the other mineral salts included KH$_2$PO$_4$, KCl, CaCl$_2$, MgSO$_4$, FeNaEDTA, H$_3$BO$_3$, MnSO$_4$-H$_2$O, ZnSO$_4$-7H$_2$O, CuSO$_4$-5H$_2$O and (NH$_4$)$_6$Mo$_7$O$_{24}$ − 4H$_2$O, supplying 137, 20.2, 152.1, 130, 31.2, 41.6, 3.23, 1.3, 0.16, 0.16, 0.1, 0.02 and 0.02 μg $g^{-1}$ dry litter for N, P, K, Ca, Mg, S, Fe, Cl, B, Mn, Zn, Cu and Mo, respectively. Mesh bags were harvested after 28 days of incubation. Each litter sample was separated into two subsamples, one of which was used to calculate the rate of oat straw decomposition, and the other to determine cellobiohydrolase activity as explained above and detailed in the S1 Appendix.

## Experiment 6. Simultaneous amendment with glucose, NH$_4$Cl and soil

The purpose of this experiment was to determine whether soil could substitute for the mineral nutrients other than N in the stimulation of oat straw decomposition. There were six replicates each of four treatments: 1) oat straw with clover litter, 2) oat straw with additional water, 3) oat straw with 500 μg C as glucose $g^{-1}$ dry litter and 137 μg N as NH$_4$Cl $g^{-1}$ dry litter, and 4) oat straw with 500 μg C as glucose $g^{-1}$ dry litter and 137 μg N as NH$_4$Cl $g^{-1}$ dry litter overlying soil. The soil had been collected from a nearby agricultural field which had been planted to maize. It had previously been allowed to dry thoroughly at room temperature. Large stones and other

debris were removed, and the soil was thoroughly mixed. To eliminate any effects of live soil organisms, dry soil was treated twice in an autoclave (120˚ C, 20 min). For treatment 4, a thin layer (0.5 cm) of the sterilized soil (50 g) was placed in the petri dishes for that treatment, rewet to field capacity, then the mesh bags were placed on top of the soil. Mesh bags were harvested after 28 days of incubation. Each litter sample was separated into two subsamples, one of which was used to calculate rate of oat straw decomposition, and the other to determine cellobiohydrolase activity as explained above and detailed in the S1 Appendix.

### Experiment 7. Simultaneous amendment with $NH_4Cl$ and other mineral nutrients

We previously showed that $NH_4Cl$ alone had no significant effect on oat straw decomposition [11], but that the combination of glucose, $NH_4Cl$ and other mineral nutrients accounted for the positive effect of clover litter on oat straw decomposition. The purpose of this experiment was to determine whether mineral nutrients in the absence of glucose could account for the clover litter effect. There were eight replicates of each of four treatments: 1) oat straw with clover litter, 2) oat straw with additional water, 3) oat straw with 137 μg N as $NH_4Cl$ $g^{-1}$ dry litter, and 4) oat straw with 137 μg N as $NH_4Cl$ $g^{-1}$ dry litter and all other mineral nutrients supplied in the same ratio to N as in Hoagland solution [30]. For treatment 4, the other mineral salts were supplied in the same way as in treatment 3 of Experiment 5. Mesh bags were harvested after 28 days of incubation. The decomposition rate of oat straw samples was calculated as explained above.

### Experiment 8. Amendment with mineral nutrients other than N

The purpose of this experiment was to determine whether the addition of mineral nutrients other than N, could account for the positive effect of clover litter on oat straw decomposition. There were eight replicates of each of four treatments: 1) oat straw with clover litter, 2) oat straw with additional water, 3) oat straw with 137 μg N as $NH_4Cl$ $g^{-1}$ dry litter and all other mineral nutrients supplied in the same ratio to N as in Hoagland solution [30], and 4) oat straw with all mineral nutrients other than N supplied in the same concentrations as in treatment 3. For treatment 3 and 4, the other nutrient salts were supplied in the same way as in treatment 3 of Experiment 5. Mesh bags were harvested after 28 days of incubation. The decomposition rate of oat straw samples was calculated as explained above.

### Data analysis

For all analyses of variance of decomposition rates and cellobiohydrolase activities, data were transformed, as appropriate, to satisfy the assumptions of normality and homogeneity of variance, either by log or reciprocal transformation. Mean separations ($P < 0.05$) were accomplished using false discovery rate-protected least significant differences (FDR-LSD). In one case the analysis of variance was significant while the FDR-LSD indicated no significant differences among means. For this single case we separated means using the simple least significant difference method. All analyses and post-hoc tests were conducted in the R software environment [31].

## Results

### Litter macronutrient concentrations

The initial concentrations of N, P, K, Ca, Mg and S were higher in clover litter than in oat straw (Table 1).

**Table 1. Initial macronutrient compositions of oat straw and clover litter.** Data are means ± SEM, n = 2.

|  | N (%) | P (%) | K (%) | Ca (%) | Mg (%) | S (%) |
|---|---|---|---|---|---|---|
| **Oat straw** | 0.813±0.054 | 0.097±0.007 | 0.285±0.092 | 0.182±0.018 | 0.082±0.000 | 0.081±0.004 |
| **Clover litter** | 4.67±0.173 | 0.379±0.005 | 3.52±0.158 | 1.36±0.061 | 0.307±0.001 | 0.238±0.007 |

### Experiment 1. Amendment with water

Additional water did not have a significant effect on oat straw decomposition (one-tailed t-test, $t_8 = 0.54$, $P = 0.699$). The mean (SE) decomposition rates were 9.97 (0.74) and 9.45 (0.62) mg g$^{-1}$ d$^{-1}$ for no water and additional water treatments, respectively.

### Experiment 2. Amendment with NH$_4$Cl

Clover litter significantly increased oat straw decomposition rate, but neither low NH$_4$Cl concentration ("N" in Fig 1) nor high NH$_4$Cl concentration ("2N") significantly affected oat straw decomposition compared to oat straw alone (Table A in S1 Appendix, Fig 1). Therefore, the addition of NH$_4$Cl could not account for the positive effect of clover litter on oat straw decomposition.

### Experiment 3.1. Amendment with glucose

In experiment 3.1, the maximum cellobiohydrolase activity occurred at a glucose concentration of 500 µg C g$^{-1}$ dry litter, although it was not significantly greater than the activity with no

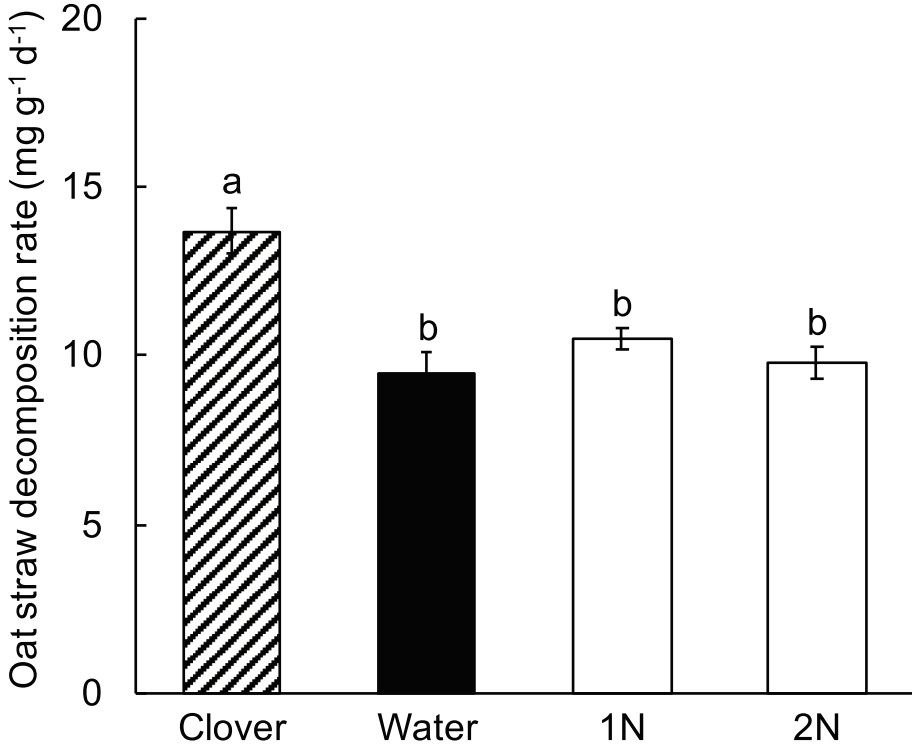

**Fig 1. Experiment 2, the effect of NH$_4$Cl amendment on decomposition of oat straw.** Clover = oat straw with clover litter; Water = oat straw with additional water; 1N = oat straw with 2000 µg N as NH$_4$Cl g$^{-1}$ dry weight oat straw; 2N = oat straw with 4000 µg N as NH$_4$Cl g$^{-1}$ dry weight oat straw. Error bars are ± 1 SEM. Different letters indicate means are significantly different according to the FDR-protected least significant difference method.

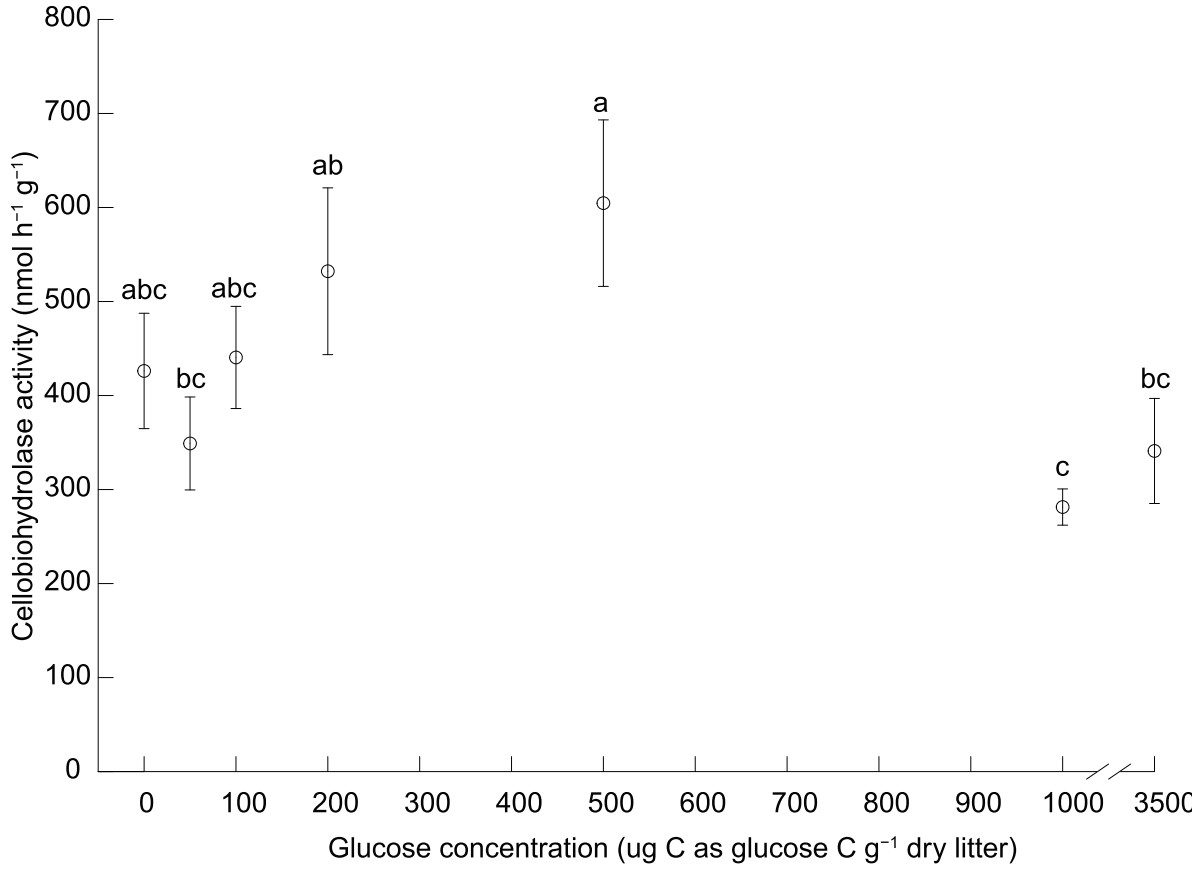

**Fig 2. Experiment 3.1, the effect of amendment of glucose on cellobiohydrolase activity in oat straw.** Open circles and error bars are means ± 1 SEM. Different letters indicate means are significantly different according to the FDR-protected least significant difference method.

glucose. Compared to the highest cellobiohydrolase activity, significantly lower activities occurred at the two highest glucose concentrations, 1000 and 3500 µg C as glucose g$^{-1}$ dry litter although, again, the activities at the two highest glucose concentrations did not differ significantly from the activity with no glucose (Table B in S1 Appendix, Fig 2).

## Relationship between cellobiohydrolase activity and oat straw decomposition

Data from experiments 3.2, 4, 5 and 6 (below) revealed a significant, positive, linear relationship between oat straw decomposition and cellobiohydrolase activity (Fig 3). Therefore, based on the cellobiohydrolase activities in Fig 2, we assumed that the optimum glucose concentration for decomposition was also 500 µg C g$^{-1}$ dry litter in subsequent experiments where glucose was added.

## Experiment 3.2. Amendment with glucose

In experiment 3.2, there was a significant positive effect of clover litter on oat straw decomposition at 28 d, but not earlier (Table C in S1 Appendix, Fig 4). At 28 d, there was also a small positive but insignificant effect of the optimum glucose concentration (500 µg C as g$^{-1}$ dry litter, designated "C" in Fig 4) on oat straw decomposition, consistent with the cellobiohydrolase

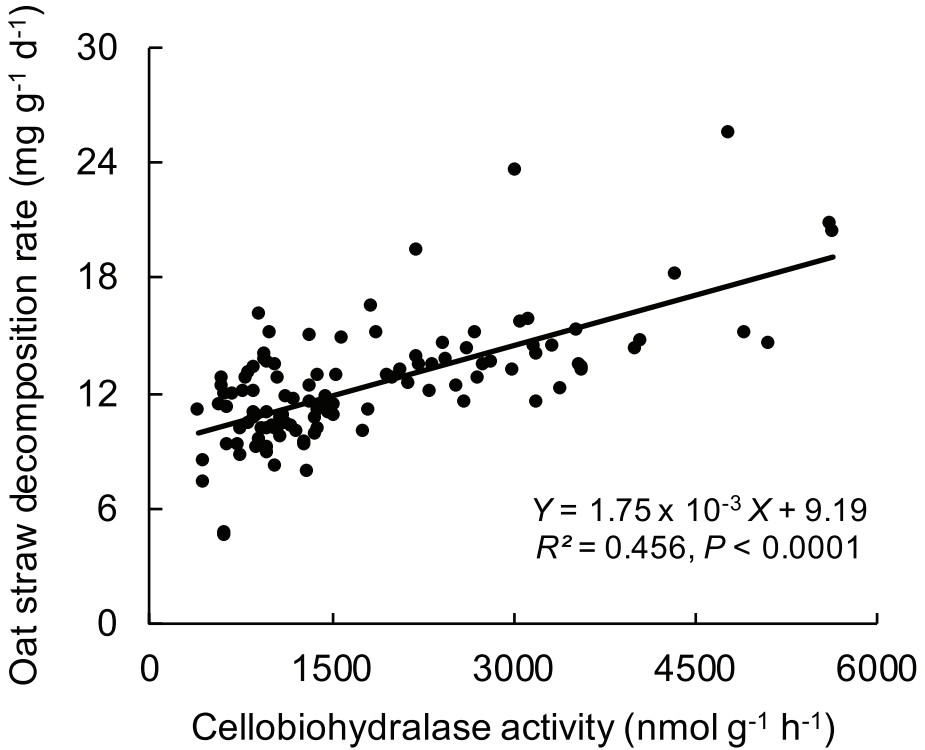

**Fig 3. The relationship between cellobiohydrolase activity and oat straw decomposition rate.** Cellobiohydrolase activities were measured at 28 d and oat straw decomposition rates were calculated between 0 and 28 days in experiments 3.2, 4, 5 and 6.

activity result of experiment 3.1. However, the high glucose concentration (4000 µg C g$^{-1}$ dry litter, designated "8C" in Fig 5) significantly reduced oat straw decomposition, and that was also consistent with the result for cellobiohydrolase activity in experiment 3.1. The significant, negative effect of the high glucose concentration first became apparent at 21 d and was even larger at 28 d.

Clover litter significantly increased cellobiohydrolase activity in oat straw relative to oat straw alone on all three sampling dates, including 14 d (Table D in S1 Appendix, Fig 4). Thus, oat straw cellobiohydrolase activity responded earlier to clover litter than did oat straw decomposition. We also found that 500 µg C as glucose g$^{-1}$ dry litter had a small but insignificant positive effect on cellobiohydrolase activity, and that 4000 µg C as glucose g$^{-1}$ dry litter significantly reduced cellobiohydrolase activity at 21 and 28 d of incubation, largely consistent with the results on oat straw decomposition, and with the cellobiohydrolase results from experiment 3.1.

### Experiment 4. Simultaneous amendment with glucose and NH$_4$Cl

Clover litter significantly increased oat straw decomposition relative to oat straw alone (Table E in S1 Appendix, Fig 5). Both the 500 µg C as glucose g$^{-1}$ dry litter treatment (designated "C" in Fig 5) and the glucose plus NH$_4$Cl treatment ("C+N") resulted in oat straw decomposition rates that were intermediate between the clover litter and control treatments. Because the C+N treatment did not increase the rate of decomposition of oat straw beyond that of glucose alone, mineral N must not have limited oat straw decomposition, consistent with the results from experiment 2.

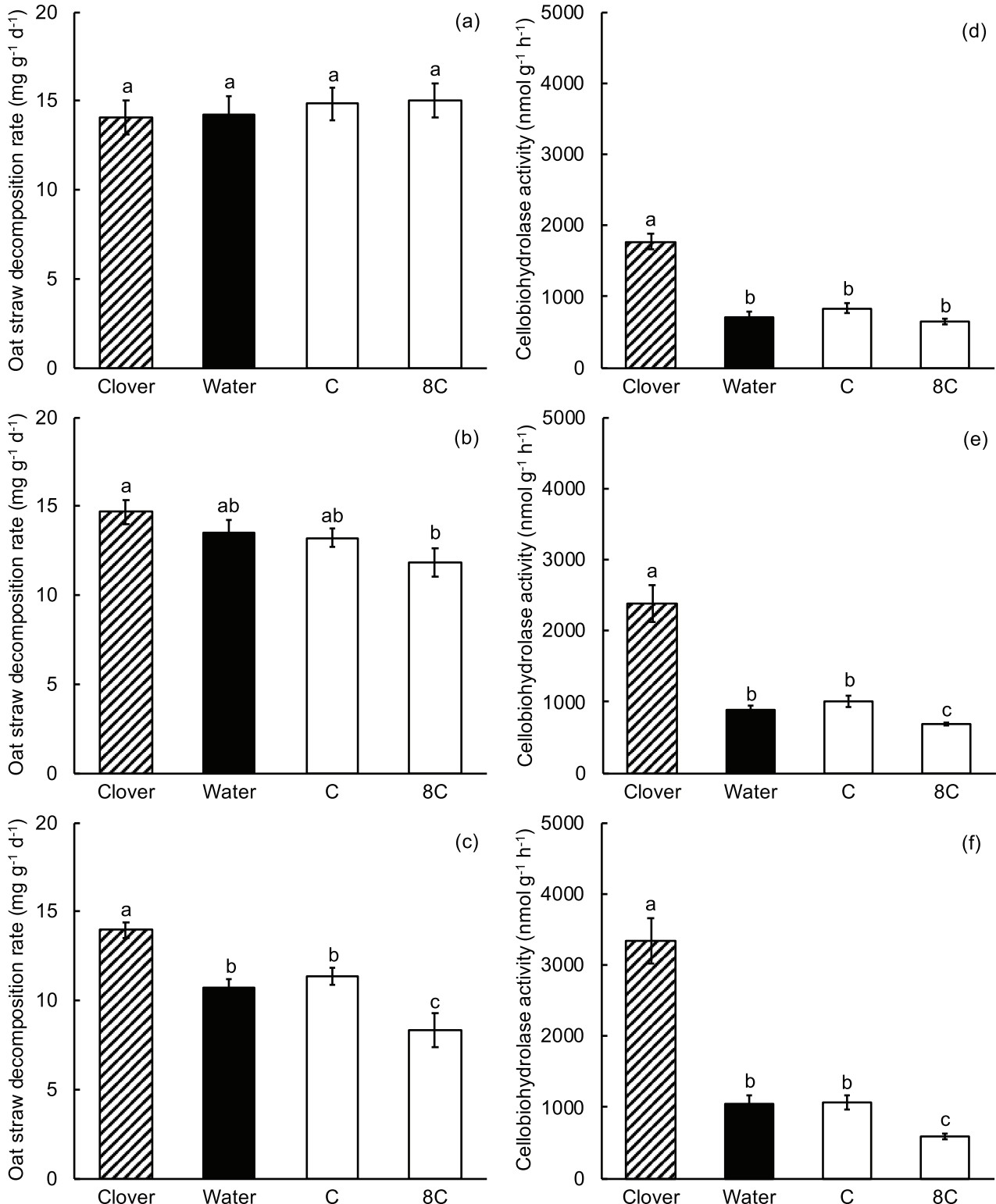

**Fig 4. Experiment 3.2, the effect of glucose amendments on oat straw decomposition rate and cellobiohydrolase activity.** 14 d (a, d), 21 d (b, e), and 28 d (c, f). Clover = oat straw with clover litter; Water = oat straw with additional water; C = oat straw with 500 μg C as glucose g$^{-1}$ dry weight oat straw; 8C = oat straw with 4000 μg C as glucose g$^{-1}$ dry weight. Error bars are ± 1 SEM. Different letters indicate means are significantly different according to the FDR-protected least significant difference method.

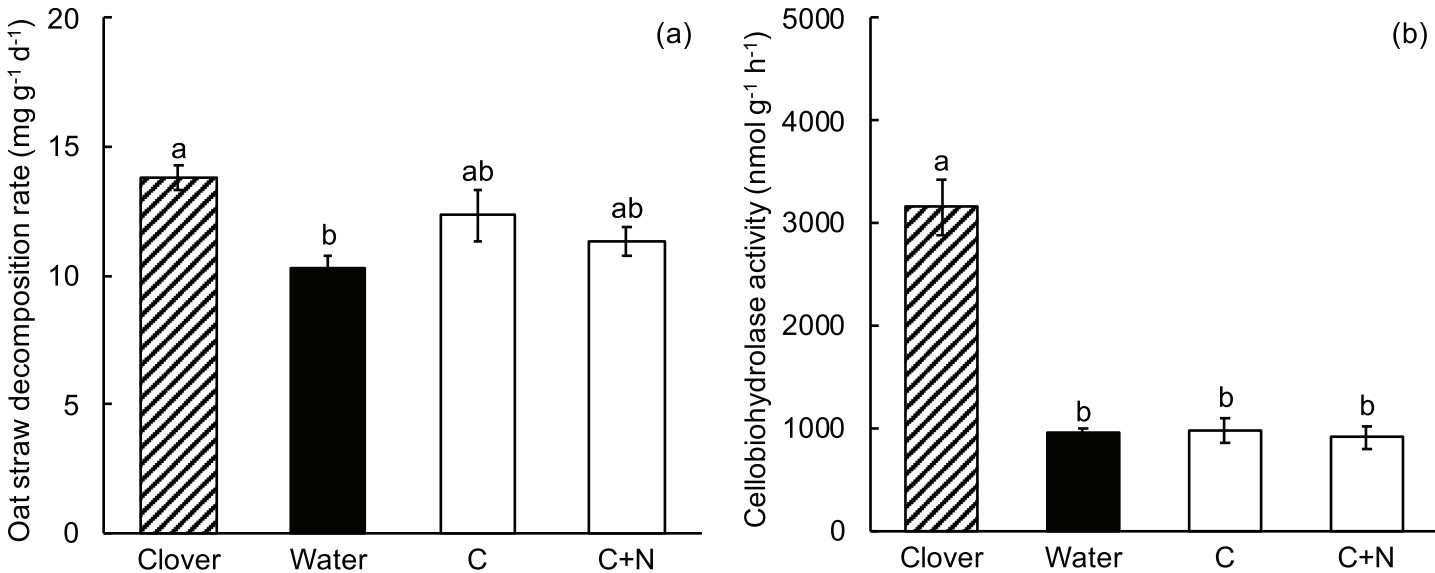

**Fig 5. Experiment 4, effect of treatment on oat straw decomposition rate (a) and cellobiohydrolase activity (b).** Clover = oat straw with clover litter; Water = oat straw with additional water; C = oat straw with 500 µg C as glucose g$^{-1}$ dry weight oat straw; C+N = oat straw with 500 µg C as glucose g$^{-1}$ dry weight oat straw and 137 µg N as NH$_4$Cl g$^{-1}$ dry weight. Error bars are ± 1 SEM. Different letters indicate means are significantly different according to the FDR-protected least significant difference method.

Clover litter also significantly increased cellobiohydrolase activity in oat straw (Table F in S1 Appendix, Fig 5). However, 500 µg C as glucose g$^{-1}$ dry litter ("C"), and glucose plus NH$_4$Cl ("C+N") did not have significant effects on cellobiohydrolase activity.

### Experiment 5. Simultaneous amendment with glucose, NH$_4$Cl and other mineral nutrients

The rate of oat straw decomposition increased significantly in the presence of clover litter (Table G in S1 Appendix, Fig 6). The simultaneous addition of glucose, NH$_4$Cl and other mineral nutrients also significantly increased oat straw decomposition rate and the resultant decomposition rate was not significantly different from that of oat straw in the presence of clover litter, indicating that some combination of C, N, and other mineral nutrients was sufficient to account for the clover litter effect.

Clover litter significantly increased cellobiohydrolase activity in oat straw relative to oat straw alone (Table H in S1 Appendix, Fig 6). Moreover, simultaneous addition of glucose, NH$_4$Cl and other mineral nutrients also significantly increased cellobiohydrolase activity, but the effect of this treatment was smaller than the effect of clover litter.

### Experiment 6. Amendment with glucose, NH$_4$Cl and soil

Clover litter significantly increased oat straw decomposition rate (Table I in S1 Appendix, Fig 7). Simultaneous amendment with glucose, NH$_4$Cl and soil also significantly increased oat straw decomposition rate compared to oat straw alone and the resultant decomposition rate was not significantly different from that of oat straw in the presence of clover litter. In other words, glucose, N and soil together were sufficient to account for the positive clover litter effect, which suggests that soil can substitute for the other mineral nutrients (other than NH$_4$Cl) that were supplied in Experiment 5.

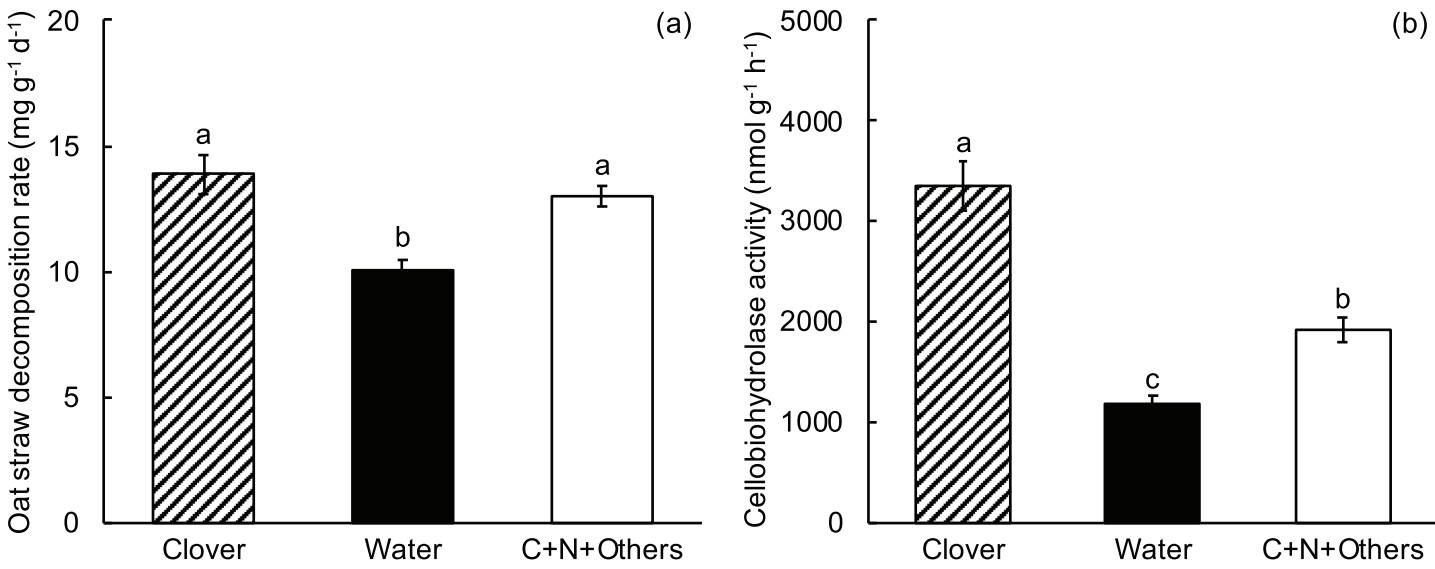

**Fig 6. Experiment 5, effect of treatment on oat straw decomposition rate (a) and cellobiohydrolase activity (b).** Clover = oat straw with clover litter; Water = oat straw with additional water; C+N+Others = oat straw with 500 μg C as glucose g$^{-1}$ dry weight oat straw, 137 μg N as NH$_4$Cl g$^{-1}$ dry weight and other mineral nutrients in a complete plant nutrient solution (see Methods). Error bars are ± 1 SEM. Different letters indicate means are significantly different according to the FDR-protected least significant difference method.

Clover litter significantly increased cellobiohydrolase activity in oat straw (Table J in S1 Appendix, Fig 7). Simultaneous addition of glucose and NH$_4$Cl did not significantly influence cellobiohydrolase activity. Simultaneous addition of glucose, NH$_4$Cl and soil significantly increased cellobiohydrolase activity, but this activity was lower than in the oat straw in association with clover litter.

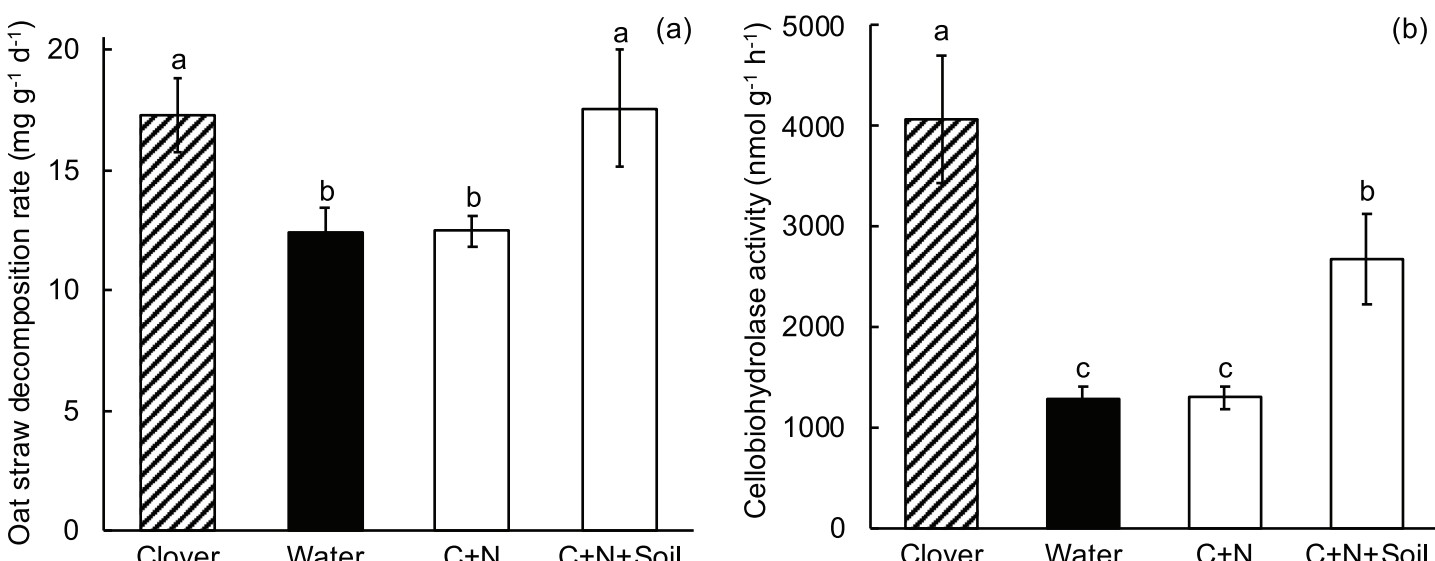

**Fig 7. Experiment 6, the effect of treatment on oat straw decomposition (a) and cellobiohydrolase activity (b).** Clover = oat straw with clover litter; Water = oat straw with additional water; C+N = oat straw with 500 μg C as glucose g$^{-1}$ dry weight oat straw and 137 μg N as NH$_4$Cl g$^{-1}$; C+N+Soil = oat straw with 500 μg C as glucose g$^{-1}$ dry weight, 137 μg N as NH$_4$Cl g$^{-1}$ dry weight, and placed on sterilized soil. Error bars are ± SEM. Different letters indicate means are significantly different according to the FDR-protected least significant difference method.

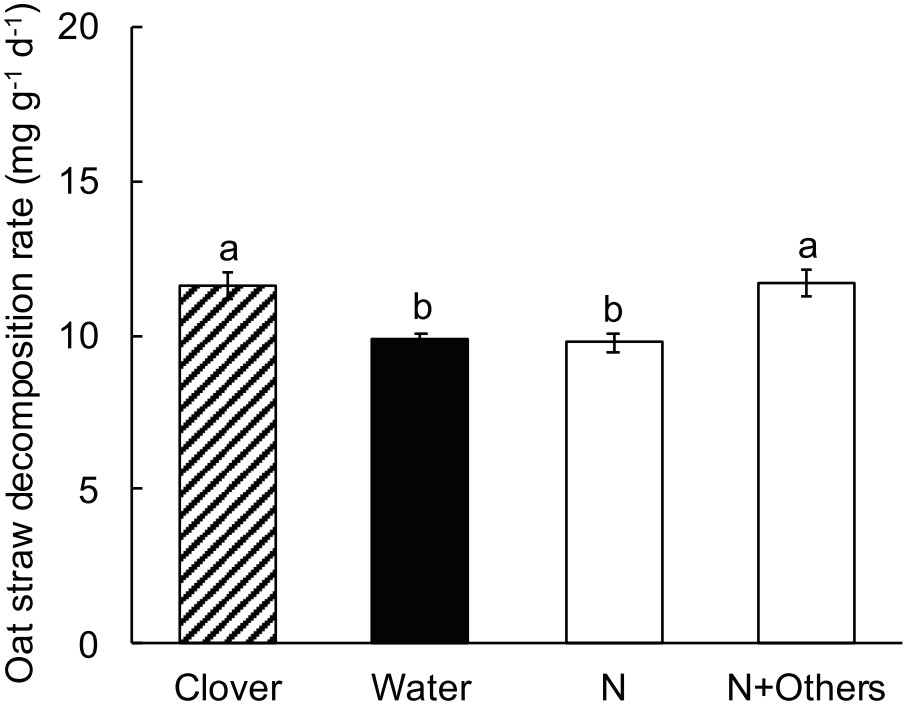

**Fig 8. Experiment 7, the effect of treatment on decomposition rate of oat straw.** Clover = oat straw with clover litter; Water = oat straw with additional water; N = oat straw with 137 μg N as $NH_4Cl$ $g^{-1}$ dry weight oat straw; N +Others = oat straw with 137 μg N as $NH_4Cl$ $g^{-1}$ dry weight and other mineral nutrients in a complete plant nutrient solution (see Methods). Error bars are ± 1 SEM. Different letters indicate means are significantly different according to the FDR-protected least significant difference method.

## Experiment 7. Simultaneous amendment with $NH_4Cl$ and other mineral nutrients

In experiment 7, oat straw decomposition increased significantly in the presence of clover litter (Table K in S1 Appendix, Fig 8). The addition of $NH_4Cl$ alone did not have a significant effect on oat straw decomposition rate, indicating that mineral N did not limit oat straw decomposition, consistent with the results from experiment 2. The simultaneous addition of $NH_4Cl$ and other mineral nutrients, however, significantly increased oat straw decomposition rate to that of oat straw with clover litter, indicating that some combination of N and other mineral nutrients limited oat straw decomposition and that this combination was sufficient to account for the clover litter effect. Therefore, the addition of glucose in experiments 5 and 6 was not necessary to account for the effects of clover litter on oat straw decomposition.

## Experiment 8. Amendment with mineral nutrients other than N

Oat straw decomposition rate increased significantly in the presence of clover litter (Table L in S1 Appendix, Fig 9). The simultaneous addition of $NH_4Cl$ and other mineral nutrients significantly increased oat straw decomposition rate compared to oat straw alone as in experiment 7. Again, because the resultant decomposition rate was not significantly different from that of oat straw in the presence of clover litter, some combination of N and one or more other mineral nutrients accounted for the clover litter effect. The addition of other mineral nutrients (in the absence of N) resulted in oat straw decomposition rates that were nearly as large, indicating that most of the stimulatory effect of clover litter or the N+Others treatment could be

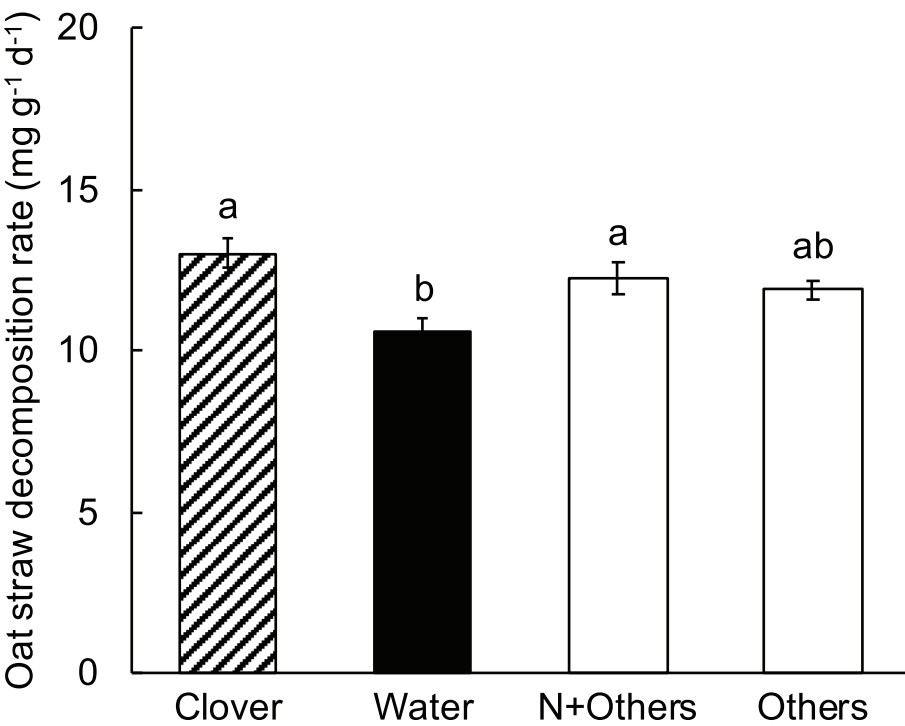

**Fig 9. Experiment 8, the effect of treatment on decomposition rate of oat straw.** Clover = oat straw with clover litter; Water = oat straw with additional water; N+Others = oat straw with 137 μg N as $NH_4Cl$ $g^{-1}$ dry weight oat straw and other mineral nutrients in a complete plant nutrient solution (see Methods); Others = oat straw with other mineral nutrients in a complete plant nutrient solution (see Methods). Error bars are ± 1 SEM. Different letters indicate means are significantly different according to the FDR-protected least significant difference method.

accounted for by mineral nutrients other than N. Thus, only when the other nutrients were added might N have become limiting.

## Discussion

We previously showed that clover litter stimulated decomposition and decomposition enzyme activity in oat straw, a case of positive, non-additive decomposition [11]. Such may be caused by the stimulation of decomposer microbe activity in one litter type by the transfer of limiting resources from another litter type, including labile C [16,17,32], water [14,15], or mineral nutrients [6,12,13,24]. We, therefore, determined whether specific resources that could be supplied to oat straw by clover litter limited decomposition enzyme activity in oat straw including water, labile C, mineral N, other mineral nutrients, or various combinations of these. Our results suggest that the transfer of water, labile N or labile C from clover litter to oat straw were not responsible for the positive effect of clover litter on oat straw decomposition. Instead, they suggest that the simultaneous transfer of N and other minerals from clover litter to oat straw caused the positive, non-additive decomposition of oat straw.

We first tested the hypothesis that the supply of water from clover litter to oat straw stimulates oat straw decomposition. Our results were not consistent with that hypothesis as additional water did not stimulate oat straw decomposition. In fact, we can envision only one circumstance in which water transfer could be responsible for positive, non-additive decomposition, which is when water is lost only from the oat straw, to be replaced only by water from the clover litter. In that case the time during which the water potential of oat straw remains favorable for decomposition could be increased by the clover litter. If, on the other hand, there

were hydraulic connectivity between soil and either litter type, the water potential decline of oat straw would be the same with or without clover litter because it would be governed by the rate of water transfer from the soil. Thus, while it is possible for water transfer from one litter type to the other to be responsible for positive, non-additive decomposition, this does not seem likely under the conditions in which we have observed positive, non-additive decomposition, namely when water loss is minimized in a high humidity enclosure.

Because clover is a legume, its tissue N concentration was much higher than that of the oat straw. We, therefore, hypothesized that clover litter enhanced oat straw decomposition by supplying mineral N, a potentially-limiting resource, to oat straw decomposer microbes. However, the addition of $NH_4Cl$ at 137, 2000 or 4000 µg N $g^{-1}$ dry weight of litter had no significant impact on oat straw decomposition. Other researchers have also shown that, in some circumstances, N transfer alone is not responsible for the positive, non-additive effect so frequently observed [33,34]. It is possible that N actually did limit microbial activity in oat straw but that $NH_4Cl$ was not a preferred N source. It is possible that organic N sources such as proteins or amino acids would have produced different results. However, the results from subsequent experiments, discussed below, suggest that N alone was not the limiting factor in oat straw decomposition and, therefore, not the reason for the positive effect of clover litter on oat straw decomposition.

We found that small concentrations of glucose (500 µg C $g^{-1}$ dry litter) could stimulate oat straw decomposition to a limited extent, but this was never statistically significant. The same concentration of glucose did significantly increase cellobiohydrolase activity, but not to the extent that clover litter did. We conclude, therefore, that clover litter did not have its primary stimulatory effect on oat straw decomposition by supplying labile C.

If N alone and glucose alone could not account for the effect of clover litter on oat straw decomposition, we felt it was possible that a combination of glucose and N might. After all, clover litter could serve as a source of both labile C and N simultaneously. However, the results from experiment 4, in which we simultaneously added glucose and $NH_4Cl$ to oat straw, did not stimulate oat straw decomposition to the extent that clover litter did.

However, our results indicated that the addition of one or more mineral nutrients to oat straw could account for the entire effect of clover litter on oat straw decomposition. When we added a combination of glucose, $NH_4Cl$ and all other mineral nutrients considered to be essential for plant growth, their effect on oat straw decomposition was indistinguishable from that of clover litter. Further, the simultaneous addition of $NH_4Cl$ and all other mineral nutrients was sufficient to account for the clover effect and, in fact, the addition of mineral nutrients other than N accounted for the majority of this. We conclude, therefore, that the positive effect of clover litter on oat straw decomposition is primarily due to the transfer of one or more mineral resources other than N. Indeed, there are occasions when P [25,26], K, Mg, Mn or Ca limit litter decomposition [28].

While soil is likely to be deficient in both labile C and N for decomposer microbe growth, it generally contains P, K, Ca, Mg and a wide range of micronutrients. Because agricultural litter is often in contact with mineral soil as it decomposes, we determined whether soil could serve as a source of mineral nutrients. We found that a soil could substitute for the mineral nutrients other than N. Thus, whether positive, non-additive decomposition occurs is not simply a function of the various litter types involved, but also the surrounding matrix, which is frequently soil.

In contrast with the optimum glucose concentration (500 µg C as glucose $g^{-1}$ dry litter) for oat straw decomposition, elevated glucose concentration (4000 µg C as glucose $g^{-1}$ dry litter) significantly retarded oat straw decomposition. Negative effects of glucose additions on decomposition has been reported previously [35,36]. This may be due to the glucose being

preferentially metabolized over the more recalcitrant litter. In other words, glucose substitutes for litter as a preferred C source for the decomposer microbes. This was supported by the low activity of cellobiohydrolase associated with the high glucose concentrations, which follows logically from the reduced need to hydrolyze cellulose.

Depressed water potential can limit microbial activity and decrease litter decomposition [37]. When glucose was added in the various experiments, therefore, litter decomposition may have been affected not only because of the additional C source, but also because of a depressed water potential. In Experiment 3.2, for example, when a total of 16,000 µg C g$^{-1}$ oat straw as glucose were added to the fiberglass pads (4,000 µg C g$^{-1}$ oat straw applied 4 times), decomposition was depressed relative to when a total of 2,000 µg C g$^{-1}$ oat straw (500 µg C g$^{-1}$ oat straw applied 4 times) were added. Assuming no glucose was consumed during the 28 days of incubation (an unlikely, worst-case scenario), 2,000 µg C g$^{-1}$ oat straw added to the fiberglass pad would be equivalent to -0.131 bars water potential, and 16,000 µg C g$^{-1}$ oat straw would be equivalent to -1.05 bar water potential. Therefore, the negative effect of the high concentration of glucose on oat straw decomposition compared to the lower concentration of glucose could have been caused by a depressed water potential and thus depressed microbial metabolism. But this seems unlikely. In an unpublished experiment of ours, a single glucose addition of 32,000 µg C g$^{-1}$ oat straw actually *increased* microbial respiration compared to a water-only control, despite reducing the water potential to approximately -2 bar, indicating that microbial metabolism is not likely impaired at the water potentials of these experiments.

We measured both oat straw decomposition and cellobiohydrolase activity in oat straw after 25 or 28 d of incubation in multiple experiments. We found a robust, positive correlation between the two, as expected. However, frequently when a treatment such as N+Others stimulated oat straw decomposition as much as did clover litter, it did not stimulate oat straw cellobiohydrolase activity to the same extent as clover litter. One should probably not expect the dynamics of hydrolytic enzymes and decomposition to be the same in different treatments. For one thing, we assessed enzyme activity as an instantaneous measure on the day of harvest, while we calculated decomposition as an integrated measure between day 0 and the day of harvest. While clover litter was capable of supplying resources continuously to decomposer microbes in oat straw, the various resources in solution (mineral nutrients, glucose) were applied only once at the beginning of the incubation or weekly throughout the incubation. Thus, the resources, at best, had been applied a week prior to the analysis of enzyme activity.

Our decomposition studies occurred over a period of 25–28 d, the earliest phase of oat straw decomposition. Positive, non-additive litter decomposition is a process that is most important in the earliest phases of decomposition [38,39]. Moreover, in experiments 7 and 8, for example, the average clover weight loss was 73.6% within 28 d. So, for this litter combination, we characterized decomposition during much of the non-additive decomposition process.

Our results are consistent with the hypothesis that clover litter can supply rate-limiting mineral nutrients to the microbes decomposing oat straw. However, because hyphae of decomposer fungi provide networks through which resources that limit microbial activity can be transferred, any material adjacent to one litter type, whether it be another type of litter or soil, could influence decomposition of the former by providing limiting resources to its decomposer organisms. Thus, in predicting litter decomposition and its consequence, including the cycling of mineral nutrients, the fluxes of C into the atmosphere, and the transformation of relatively labile litter into stable soil organic matter, it may be important to consider litter in the context of an entire decomposition system, which includes the litter and the adjacent soil connected to the litter by hyphae of decomposer fungi. Indeed, we found that soil could supply to oat straw rate-limiting minerals except N. Failure to account for soil in the

decomposition system may explain why positive, non-additive decomposition is observed in some studies and not in others.

## Supporting information

**S1 Appendix.**
(DOCX)

## Acknowledgments

We thank Mitchell C. Hunter for help procuring the litter used in this study.

## Author Contributions

**Conceptualization:** Na Yin, Roger T. Koide.

**Data curation:** Na Yin, Roger T. Koide.

**Formal analysis:** Na Yin.

**Funding acquisition:** Roger T. Koide.

**Investigation:** Na Yin, Roger T. Koide.

**Methodology:** Roger T. Koide.

**Project administration:** Roger T. Koide.

**Resources:** Roger T. Koide.

**Supervision:** Roger T. Koide.

**Validation:** Roger T. Koide.

**Visualization:** Na Yin.

**Writing – original draft:** Roger T. Koide.

**Writing – review & editing:** Na Yin, Roger T. Koide.

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
