## [Decision Letter · Decision Letter 0]

30 Oct 2019

PONE-D-19-21163

The role of resource transfer in positive, non-additive litter decomposition

PLOS ONE

Dear Dr. Koide,

Thank you for submitting your manuscript to PLOS ONE. After careful consideration, we feel that it has merit but does not fully meet PLOS ONE’s publication criteria as it currently stands. Therefore, we invite you to submit a revised version of the manuscript that addresses the points raised during the review process.

We would appreciate receiving your revised manuscript by Dec 14 2019 11:59PM. To enhance the reproducibility of your results, we recommend that if applicable you deposit your laboratory protocols in protocols.io, where a protocol can be assigned its own identifier (DOI) such that it can be cited independently in the future. For instructions see: http://journals.plos.org/plosone/s/submission-guidelines#loc-laboratory-protocols

We look forward to receiving your revised manuscript.

Kind regards,

Shuijin Hu, PhD

Academic Editor

PLOS ONE

Journal Requirements:

"None".

Additional Editor Comments (if provided):

Both reviewers have some good comments and suggestions. Please carefully address them.

Reviewers' comments:

Reviewer's Responses to Questions

**Comments to the Author**

1. Is the manuscript technically sound, and do the data support the conclusions?

Reviewer #1: Partly

Reviewer #2: Yes

2. Has the statistical analysis been performed appropriately and rigorously? 

Reviewer #1: Yes

Reviewer #2: Yes

3. Have the authors made all data underlying the findings in their manuscript fully available?

Reviewer #1: Yes

Reviewer #2: Yes

4. Is the manuscript presented in an intelligible fashion and written in standard English?

Reviewer #1: Yes

Reviewer #2: Yes

5. Review Comments to the Author

Reviewer #1: Major comments:

The manuscript in consideration aims to explore the mechanisms to explain the positive effect of clover litter on oat decomposition. The authors conducted several independent but complementary experiments to address the mechanisms, which seem methodologically sound. It is very interesting and the manuscript is clear. However, there are several concerns need to be solved before consider publication.

1. I suggest the authors to give clear hypotheses they would like to test in the Introduction;

2. The major concern I have is that all the experiments can only explain the effect of water, N, C, other mineral nutrients or various combinations of these on oat decomposition, rather than the positive effect of clover litter on oat decomposition, which the authors want to address;

3. How the authors exclude the role of microbes from clover litter in stimulating oat straw decomposition.

Specific Comments:

• Line 46, please define C, where it first appeared in the Text

• Line 85, more references should be provided

• Line 291, “addition” typo

• Line 459-471, no studies were cited

•

Reviewer #2: Please see the attached document to view my editing suggestions. The manuscript is well-written and covers an important topic. The authors did a fine job in testing multiple treatments to better understand the effects of clover residues on oat straw decomposition. My primary editing suggesting is that the authors spend more time discussing the practical implications of their findings. Clover and oat are common cover crops and often used in mixtures. Findings from this study have practical implications for N management on subsequent crops.

6. PLOS authors have the option to publish the peer review history of their article (what does this mean?). If published, this will include your full peer review and any attached files.

Reviewer #1: No

Reviewer #2: No

---

## [Author Response · Author response to Decision Letter 0]

31 Oct 2019

We have revised the manuscript in accordance with PLOS ONE style templates. Moreover, we have responded to the following reviewer comments in the following ways:

Reviewer #1: Major comments:

1. I suggest the authors to give clear hypotheses they would like to test in the Introduction;

We have added explicit hypotheses.

2. The major concern I have is that all the experiments can only explain the effect of water, N, C, other mineral nutrients or various combinations of these on oat decomposition, rather than the positive effect of clover litter on oat decomposition, which the authors want to address;

In fact, we did address the positive effect of clover litter on oat decomposition. We had several hypotheses to explain the clover effect, including its contribution to oat straw of water, N, C and other mineral nutrients. 

3. How the authors exclude the role of microbes from clover litter in stimulating oat straw decomposition.

We have explained that in a previously published paper, the positive net effect of clover litter on oat straw decomposition was correlated with increased microbial activity. In this particular manuscript, we have extended our previous finding to the mechanisms responsible for the microbial effect. 

Specific Comments:

• Line 46, please define C, where it first appeared in the Text

This has been added.

• Line 85, more references should be provided

There are already 3 references for this single point. We do not feel it is necessary to add more to make the point.

• Line 291, “addition” typo

This has been corrected.

• Line 459-471, no studies were cited

We have added a citation here.

Reviewer #2: Please see the attached document to view my editing suggestions. The manuscript is well-written and covers an important topic. The authors did a fine job in testing multiple treatments to better understand the effects of clover residues on oat straw decomposition. My primary editing suggesting is that the authors spend more time discussing the practical implications of their findings. Clover and oat are common cover crops and often used in mixtures. Findings from this study have practical implications for N management on subsequent crops.

We have altered the manuscript as suggested by this reviewer in the manuscript.

However, we have not added further discussion on practical implications for N management on subsequent crops because the context of this work is larger than N management in agricultural systems. For example, the work has implications with respect to the cycling of mineral nutrients, the fluxes of C into the atmosphere, and the transformation of relatively labile litter into stable soil organic matter. We have added these considerations into the discussion.

---

## [Editor Report · Decision Letter 1]

4 Nov 2019

The role of resource transfer in positive, non-additive litter decomposition

PONE-D-19-21163R1

Dear Dr. Koide,

We are pleased to inform you that your manuscript has been judged scientifically suitable for publication and will be formally accepted for publication once it complies with all outstanding technical requirements.

With kind regards,

Shuijin Hu, PhD

Academic Editor

PLOS ONE

Additional Editor Comments (optional):

Dear Roger,

Congratulations! Your paper is very interesting!

Shuijin
---

## [Editor Report · Acceptance letter]

8 Nov 2019

PONE-D-19-21163R1 

The role of resource transfer in positive, non-additive litter decomposition 

Dear Dr. Koide:

I am pleased to inform you that your manuscript has been deemed suitable for publication in PLOS ONE. Congratulations! Your manuscript is now with our production department. 

With kind regards,

on behalf of

Professor Shuijin Hu 

Academic Editor

PLOS ONE